# Ensembles of Classifiers: a Bias-Variance Perspective

**Neha Gupta**[*]                                               *nehagupta@cs.stanford.edu*
*Stanford University*

**Jamie Smith**                                                *jamieas@google.com*
*Google*

**Ben Adlam**                                                  *adlam@google.com*
*Google*

**Zelda Mariet**                                               *zmariet@google.com*
*Google*

**Reviewed on OpenReview:** *https://openreview.net/forum?id=lIOQFVncY9*

## Abstract

Ensembles are a straightforward, remarkably effective method for improving the accuracy, calibration, and robustness of neural networks on classification tasks. Yet, the reasons underlying their success remain an active area of research. Building upon Pfau (2013), we turn to the bias-variance decomposition of Bregman divergences in order to gain insight into the behavior of ensembles under classification losses. Introducing a dual reparameterization of the bias-variance decomposition, we first derive generalized laws of total expectation and variance, then discuss how bias and variance terms can be estimated empirically. Next, we show that the dual reparameterization naturally introduces a way of constructing ensembles which reduces the variance and leaves the bias unchanged. Conversely, we show that ensembles that directly average model outputs can arbitrarily increase or decrease the bias. Empirically, we see that such ensembles of neural networks may *reduce* the bias. We conclude with an empirical analysis of ensembles over neural network architecture hyperparameters, revealing that these techniques allow for more efficient bias reduction than standard ensembles.

## 1 Introduction

The success of deep learning has necessitated many extensions of classical learning theory to modern models and algorithms. In particular, the double-descent phenomenon (Belkin et al., 2019) has inspired a wide breadth of research, in which the classical bias-variance decomposition (BVD) has been key (Neal et al., 2019; Adlam & Pennington, 2020). Such analyses have been instrumental to understanding the performance of various strategies that improve deep learning methods, including ensembles of deep networks (Hansen & Salamon, 1990; Lakshminarayanan et al., 2017)—a simple method with state-of-the-art robustness and uncertainty results (Ovadia et al., 2019; Gustafsson et al., 2020).

Most bias-variance analyses are specific to the mean-squared error (MSE). Although one can analyze classifiers from an MSE perspective (Yang et al., 2020), such a restriction inevitably reduces the power of the analysis. However, generalizing bias-variance decompositions to non-MSE losses is challenging. Although the MSE allows a decomposition based on the performance of the mean predictor, this is a peculiarity of the MSE rather than the rule. In the general case, bias-variance decompositions require manipulating a "central prediction" that is much less amenable to analysis. Although bias-variance decompositions exist for Bregman divergences (including the KL divergence from which standard classification losses are derived) (Pfau, 2013), such decompositions are difficult to interpret and appear to break away from standard intuitions.

---

[*]Work done at Google.

We begin by bridging the gap between the classical bias-variance decomposition and its generalization to Bregman divergences by means of a dual reparameterization. Analyzing the properties of the bias and variance for non-symmetric losses, we characterize their departure from standard intuition. In particular, we will see that ensembles that average model outputs can either increase or decrease the bias under non-symmetric losses, whereas the bias is unchanged under the MSE. We then further leverage the dual reparameterization to show that alternate ensembling techniques can recover the expected behavior of ensembles under the MSE.

We then investigate the empirical behavior of ensembles of neural networks. We first contrast the behaviors of the two natural ensembling choices that arise from the generalized BVD. We find that, in practice, ensembles that average model outputs *decrease* the bias, and discuss the conditions under which one ensemble construction is preferable to the other. Hypothesizing that bias reduction is key to the performance of recent ensembling techniques, we finally evaluate ensembles of networks of different architectures, showing that the resulting bias reduction is indeed more efficient than that of ensembles of networks of identical architectures.

**Contributions.** Using a dual reparameterization of the central prediction in the bias-variance decomposition of Pfau (2013), we tease apart which behaviors of the bias and variance are specific to the mean squared error,[1] and which behaviors are common to all Bregman divergences. Our key contributions are the following.

- The key quantity (the "central prediction") in the BVD of a Bregman divergence is the *primal* projection of the expected *dual* prediction; primal and dual spaces are specific to the choice of Bregman divergence.

- This central prediction follows a generalized law of total expectation. Similarly, the resulting variance in the BVD of a Bregman divergence satisfies a generalized law of total variance.

- Conditional estimates of the bias and variance are biased by an irreducible quantity which overestimates the bias and underestimates the variance; iterative bootstrapping can improve these estimates.

- We (re)define primal and dual ensembling methods based on the BVD. For the cross-entropy loss, primal ensembling averages model probabilities, whereas dual ensembling averages model logits.

- Dual ensembling recovers the standard properties of ensembles under the MSE loss (unchanged bias and reduced variance). However, ensembling in primal space can either increase or decrease the bias.

- Empirically, we see on Cifar10 and Cifar100 that primal ensembling neural networks under the cross-entropy loss tends to *decrease* the bias, achieving a lower loss than dual ensembling. This can be explained by the larger influence of extreme predictions under dual ensembling.

- Bias reduction contributes to the improved performance of ensembles of neural networks of different architectures; such ensembles reduce the bias more efficiently as a function of ensemble size.

## 2 Related work

**Bias-variance decompositions.** The bias-variance decomposition has been an important tool in understanding the behavior of machine learning models (Geman et al., 1992; Kong & Dietterich, 1995; Breiman, 1996; Adlam & Pennington, 2020; Yang et al., 2020; d'Ascoli et al., 2020; Neal et al., 2019; Lin & Dobriban, 2021). Key to the decomposition are the notions of a "central label" and of a "central prediction." Most analyses focus on the bias-variance decomposition for the Euclidean square loss. In this case, the central label and prediction correspond respectively to the *mean* label and prediction. James (2003) proposed a more general decomposition for symmetric losses, Domingos (2000) focussed on the 0-1 loss, and Pfau (2013) proposed a generalization to the space of all Bregman divergences. Hansen & Heskes (2000) identify which loss functions admit bias-variance decompositions with specific properties (but do not analyze the decompositions themselves). Jiang et al. (2017); Buschjäger et al. (2020) decompose twice differentiable losses via second order Taylor expansion. In the general case—including for the KL divergence—the resulting decompositions are approximate. Additionally, the resulting variance term may depend on target labels (Buschjäger et al., 2020, via the loss $l(\mu)$ in Eqs. 1, 2), in a significant departure from other standard definitions of a variance.

---

[1]Or, more accurately, to *symmetric* losses.

**Bregman divergences and KL divergence.** Bregman divergences are a generalization of the notion of distance, similar to but less restrictive than metrics. Bregman divergences and operations in their associated dual space are instrumental to optimization techniques such as mirror descent and dual averaging (Nemirovski & Yudin, 1983; Nesterov, 2009; Juditsky et al., 2021). Closer to our work, the Bregman representative defined in Banerjee et al. (2005) is related to the central label defined in (Pfau, 2013). The KL divergence and mean squared error are the most commonly used Bregman divergences in machine learning. The Itakura-Saito distance appears in matrix factorization tasks for audio processing (Févotte et al., 2009; Lefevre et al., 2011). Recent work has also looked at parametric (Amid et al., 2019) or learned Bregman divergences (Cilingir et al., 2020; Siahkamari et al., 2020; Lu et al., 2022).

Due to its importance in machine learning, the KL divergence is one of the few non-MSE Bregman divergences for which the bias-variance tradeoff has been specifically analyzed (Heskes, 1998; Yang et al., 2020). For the KL divergence, the "central predictor" corresponds to an average in log-probability space, which has been studied in many works, including (Brofos & Shu, 2019; Webb et al., 2020). The fact that the bias remains unchanged when averaging log-probability space has been briefly mentioned in (Dietterich, 2005).

**Ensembles of deep networks.** Ensembles combine the predictions of multiple models to improve upon the performance of a single model; see, *e.g.*, (Zhou, 2019; Dietterich, 2000). Recently, ensembles of neural networks that only differ in their random seed ("deep ensembles") have been shown to be a particularly strong baseline for a variety of benchmarks (Lakshminarayanan et al., 2017). This result prompted further research into alternative ensemble methods, including ensembles over hyper-parameters (Wenzel et al., 2020), architectures (Zaidi et al., 2021), and joint ensemble training (Webb et al., 2020). In parallel, several hypotheses have been proposed to explain the performance of deep ensembles. Fort et al. (2019) showed empirically that deep ensembles explore diverse modes in the loss landscape. Allen-Zhu & Li (2020) argued that the effectiveness of deep ensembles hinges on the assumption that inputs to the model have multiple correct features that are learned by the different ensemble members. Wilson & Izmailov (2020); Hoffmann & Elster (2021) analyzed deep ensembles as a Bayesian averaging procedure. Masegosa (2020); Ortega et al. (2021) focus on understanding the relationship between generalization and diversity for deep ensembles, both from theoretical and empirical perspectives. Lobacheva et al. (2020); Kobayashi et al. (2021) considered the interplay between model and ensemble size.

## 3 Bias-Variance decomposition

We begin with some background on Bregman divergences, which also serves to set our notation. We refer to Cesa-Bianchi & Lugosi (2006) for further background on Bregman divergences.

Although our empirical analysis focuses on the specific case of the KL divergence, all theoretical results are stated for arbitrary Bregman divergences; proofs not provided in the main text can be found in Appendix A.

### 3.1 Preliminaries

Let $\mathcal{X}$ be a closed, convex subset of $\mathbb{R}^d$, and let $F : \mathcal{X} \to \mathbb{R}$ be a strictly convex, differentiable function over $\mathcal{X}$. The Bregman divergence associated with $F$ is the function $D_F : \mathcal{X} \times \mathcal{X} \to \mathbb{R}^+$, such that

$$D_F[y\|x] := F(y) - F(x) - \nabla F(x)^\top (y - x). \tag{1}$$

It follows directly from the convexity of $F$ that $D_F$ is convex in its first argument, although not necessarily in its second argument (Bauschke & Borwein, 2001).

*Example* 3.1. The Bregman divergence of the negative entropy $F(x) = \sum_i x_i \log x_i$ is the generalized KL divergence $D_F[y\|x] = \mathrm{KL}[y\|x] - \sum_i (y_i - x_i)$. For $x$ and $y$ in the probability simplex, $D_F[y\|x] = \mathrm{KL}[y\|x]$.

The Bregman divergence of the convex conjugate $F^*$ of $F$ is also of particular importance to our analysis. We recall that the convex conjugate of a convex function $F$ is defined as

$$F^*(z) = \sup_x \ \langle z, x \rangle - F(x).$$

*Example* 3.2. The convex conjugate of the negative entropy function on the probability simplex is the log-sum-exp function (Boyd & Vandenberghe, 2004, Example 3.25).

As we assume that $F$ is differentiable, we denote by $x^* = \nabla F(x)$ the *dual* of $x \in \mathcal{X}$. We will require the following identity between primal $x$ and dual $x^*$:

$$x = (x^*)^* = \nabla F^*(\nabla F(x)). \tag{2}$$

*Example* 3.3. Let $F$ be the negative entropy over the probability simplex, whose convex conjugate $F^*$ is the log-sum-exp function. Then, $\nabla F^*$ is the softmax function, and Equation (2) yields the identity $x = \text{softmax}(\log x - 1) = \text{softmax} \log x$, where we let log operate element-wise on the vector $x$.

A Bregman divergence $D_F$ and its conjugate equivalent $D_{F^*}$ are related by the following equality:

**Proposition 3.1** (Cesa-Bianchi & Lugosi (2006)). $\forall x, y \in \mathcal{X}, D_F[x \parallel y] = D_{F^*}[y^* \parallel x^*]$.

*Example* 3.4. Let $F$ be the negative entropy over the probability simplex, and write log-sum-exp as LSE.

$$D_{F^*}[y^* \| x^*] = \text{LSE}(y^*) - \text{LSE}(x^*) - \text{softmax}(x^*)^\top (y^* - x^*).$$

## 3.2 General statement

Let $X$ be a random variable over $\mathcal{X} \subseteq \mathbb{R}^d$ representing predictions made by a machine learning model, and $Y$ be the random variable associated with the corresponding label. Randomness in $X$ may come from the choice of training data, the seed used to initialize the training algorithm, or any other source of randomness in the process that is used to obtain the model generating predictions. In contrast, randomness in $Y$ is typically due to aleatoric uncertainty, such as sensor noise.

A bias-variance analysis decomposes the average divergence $\mathbb{E}D[Y\|X]$ between label and prediction variables into a bias and two separate variance terms. The bias measures the divergence between the average label and the average prediction. The variances measure the amount of fluctuation in the labels (Bayes error) and predictions (model variance). These fluctuations are measured around an "average" or "central" point.

Under the mean squared error, these central points are respectively the expected label and expected prediction. More generally, for an arbitrary Bregman divergence, these central labels and predictions are defined as minimizers of the expected Bregman divergence with respect to the corresponding random variable.

**Definition 3.5** (Central label). Let $Y$ be a random variable over $\mathcal{X}$ (intuitively, the label). We call the unique minimizer $\arg\min_{z \in \mathcal{X}} \mathbb{E}D[Y\|z]$ the *central label*.

**Definition 3.6** (Central prediction). Let $X$ be a random variable over $\mathcal{X}$ (intuitively, the prediction). We call the unique minimizer $\arg\min_{z \in \mathcal{X}} \mathbb{E}D[z\|X]$ the *central prediction*.

**Proposition 3.2** (Banerjee et al. (2005)). *The central label satisfies* $\arg\min_{z \in \mathcal{X}} \mathbb{E}D[Y\|z] = \mathbb{E}Y$.

By analogy to Proposition 3.2, and for reasons that will be clear momentarily, we refer to the minimizer $z = \arg\min_z \mathbb{E}D[z\|X]$ as the *dual mean*, and write it $\mathcal{E}X$.

We can now write out the bias-variance decomposition for any Bregman divergence $D$ (Pfau, 2013):

$$
\begin{aligned}
\mathbb{E}D[Y\|X] &= \mathbb{E}D[Y\|\mathbb{E}Y] & \textit{(Bayes error)} \\
&+ D[\mathbb{E}Y\|\mathcal{E}X] & \textit{(bias)} \\
&+ \mathbb{E}D[\mathcal{E}X\|X]. & \textit{(model variance)}
\end{aligned} \tag{3}
$$

Because Bregman divergences are necessarily symmetric, Equation (3) takes on a more complicated form than the mean squared error's decomposition. In particular, the central prediction is no longer the expected prediction (whereas the central label remains unchanged); additionally, the ordering of terms within the bias and variances is now meaningful.

A main obstacle in bias-variance decompositions of Bregman divergences lies in the unwieldy form of the central prediction, $\mathcal{E}X = \arg\min_z \mathbb{E}D[z \parallel X]$. Our first contribution resolves this difficulty via a simple observation: much like the central label, the central prediction is also the expectation of a random variable; only, this expectation is taken in the dual space defined by the convex function $F$.

**Proposition 3.3** (Dual mean). *The dual mean $\mathcal{E}X$ is the primal projection of the mean of $X$ in dual space:*

$$\mathcal{E}X = (\mathbb{E}X^*)^* = \nabla F^*(\mathbb{E}\nabla F(X)).$$

*Proof.* By simple application of Propositions 3.1 and 3.2.

$$\arg\min_{z\in\mathcal{X}} \mathbb{E}D_F[z \parallel X] = \arg\min_{z\in\mathcal{X}} \mathbb{E}D_{F^*}[X^* \parallel z^*] = \left( \arg\min_{z^*\in\mathcal{X}^*} \mathbb{E}D_{F^*}[X^* \parallel z^*] \right)^* = (\mathbb{E}X^*)^*. \qquad \square$$

This reformulation is crucial to our analysis, and, to the extent of our knowledge, novel.[2]

*Remark* 3.7. When $D_F$ is symmetric, it follows from Proposition 3.2 that $\mathcal{E}X = \mathbb{E}X$.

*Example* 3.8. When $F$ is the negative entropy over the probability simplex, $D_F$ is the KL divergence, and

$$\mathcal{E}X = \text{softmax}(\mathbb{E}\log X).$$

### 3.3 Laws of total expectations and variance

Despite its more general form, the model variance $\mathbb{E}D[\mathcal{E}X \parallel X]$ of Equation (3) satisfies fundamental properties associated with the usual variance $\mathbb{V}X = \mathbb{E}(X - \mathbb{E}X)^2$. In particular, denoting the operation $X \to \mathbb{E}D[\mathcal{E}X \parallel X]$ by $\mathcal{V}$, one can easily verify that $\mathcal{V}X \geq 0$, and that $\mathcal{V}X = 0$ if and only if $X$ is almost surely constant.

We next show that $\mathcal{V}X$ also follows a generalization of the law of total variance. This law disentangles the effect of different sources of randomness, and is thus a fundamental tool in model analysis. Given two variables $X$ and $Z$, the law of total variance decomposes the standard (Euclidean) variance as $\mathbb{V}X = \mathbb{E}[\mathbb{V}(X \mid Z)] + \mathbb{V}[\mathbb{E}(X \mid Z)]$: the variance of $X$ is the sum of the variances respectively *unexplained* and *explained* by $Z$.

We begin by showing that the dual mean satisfies its own form of the law of total expectation; proving this is straightforward with the reparameterization of Proposition 3.3.

**Lemma 3.1.** *Let $X, Z$ be random variables on $\mathcal{X}$. Then $\mathcal{E}X = \mathcal{E}_Z[\mathcal{E}[X|Z]]$, where $\mathcal{E}[X \mid Z] := (\mathbb{E}[X^*|Z])^*$.*[3]

*Proof.* The proof follows directly from the (standard) law of total expectation and Equation (2).

$$\mathcal{E}(\mathcal{E}[X|Z]) = \mathcal{E}\left[ (\mathbb{E}[X^*|Z])^* \right] = \left( \mathbb{E}\left[ \mathbb{E}[X^*|Z] \right] \right)^* = (\mathbb{E}X^*)^* = \mathcal{E}X. \qquad \square$$

With Lemma 3.1 in hand, we can easily show a generalized form of the law of total variance, which simply accounts for the definition of dual mean and generalized variance.

**Lemma 3.2.** *Let $X, Z$ be random variables over $\mathcal{X}$, and define the conditional variance $\mathcal{V}[X|Z] := \mathbb{E}\big[D[\mathcal{E}(X|Z) \parallel X]\big|Z\big]$. The variance $\mathcal{V}X := \mathbb{E}D[\mathcal{E}X\|X]$ satisfies a generalized law of total variance:*

$$\mathcal{V}[X] = \mathbb{E}[\mathcal{V}[X|Z]] + \mathcal{V}[\mathcal{E}[X|Z]].$$

As above, we define the conditional variance by replacing $\mathbb{E}[\cdot]$ with $\mathbb{E}[\cdot|Z]$. We cannot overstate the importance of Lemma 3.2, which is key to disentangling sources of randomness in ML algorithms (Neal et al., 2019; Adlam & Pennington, 2020; Lin & Dobriban, 2021).

---

[2]Pfau (2013) showed that $z = \arg\min_z \mathbb{E}D[z\|X]$ satisfies $\nabla F(z) = \mathbb{E}[\nabla F(X)]$, which also entails Prop. 3.3.
[3]This definition is simply the definition of $\mathcal{E}$ using the conditional expectation $\mathbb{E}[\cdot|Z]$ in place of $\mathbb{E}[\cdot]$.

## 4 Conditional and bootstrapped estimates

Before turning our attention to ensembles, we briefly discuss empirical estimates of the bias and variance.

The bias-variance decomposition of Equation (3) applies to any source of randomness. This includes the random seed chosen to train the model, as well as the randomness in the training data $T$. However, although one can easily draw a new random seed and retrain a model, sampling a new training set is often more difficult. Even when acquiring more data is possible, this data is often used to augment an existing training set rather than to evaluate the model's robustness to data randomness.

In settings where there is randomness that cannot be controlled for, the empirical estimates of the bias and variance of Equation (3) will necessarily be only approximate. For example, if we only get one draw of the training set, our estimates will be *conditioned* on the available training data. The following proposition quantifies to which extent these conditional quantities depart from their unconditional equivalents.

**Proposition 4.1.** *Let $X, Z$ be two random variables over $\mathcal{X}$; to simplify notation, we assume that the label $Y$ is deterministic ($Y = y \in \mathcal{X}$). Applying Equation (3) to $X|Z$ then taking the expectations over $Z$ yields an alternate BVD:*

$$\mathbb{E}D[y\|X] = \underbrace{\mathbb{E}_Z D[y\|\mathcal{E}(X|Z)]}_{\text{Conditional bias: Bias}_Z} + \underbrace{\mathbb{E}_Z \mathbb{E}\Big[D[\mathcal{E}(X|Z)\|X]\Big|Z\Big]}_{\text{Conditional variance: Var}_Z}.$$

*The conditional bias (resp. variance) overestimates (resp. underestimates) their respective total values by the fixed quantity $\mathbb{E}_Z D[\mathcal{E}X\|\mathcal{E}(X|Z)]$:*

$$\text{Bias}_Z = \text{total bias} + \mathbb{E}_Z D[\mathcal{E}X\|\mathcal{E}(X|Z)] \qquad \text{Var}_Z = \text{total variance} - \mathbb{E}_Z D[\mathcal{E}X\|\mathcal{E}(X|Z)].$$

*Remark* 4.1. The quantity $\mathbb{E}_Z D[\mathcal{E}X\|\mathcal{E}(X|Z)]$ is non-negative, and equal to 0 if $X$ and $Z$ are independent.

Alternatively, one can estimate the true bias and variance by partitioning the training set into disjoint subsets; models are trained on these subsets, which act as different draws of the training distribution. This approach yields an unbiased estimator that is consistent as the number of partitions goes to infinity.[4] However, partitioning raises the difficult question of how (and if) the model should also be modified: for a neural network, should the width or number of layers be reduced to accommodate the smaller training set? Furthermore, this approach is by construction incapable of estimating the bias and variance of models trained on the full training set.

A second option lies in bootstrapping (Efron & Tibshirani, 1994; Hall, 1992): we create new datasets by sampling with replacement from the original dataset, and use these samples to estimate the bias and variance. Repeating this procedure by *re*sampling from the bootstrap samples in turn estimates the quantity required to correct (Hall, 1992) the bootstrapped estimate (Algorithm 1 and Figure 2). This approach does not require reducing the size of the training set, but bootstrapped samples will contain duplicate points.

To compare the conditional and bootstrapped estimates, we trained Wide ResNets (WRNs) (Zagoruyko & Komodakis, 2016) with the cross-entropy loss on different disjoint partitionings of the CIFAR-10 dataset. Figure 1a uses 50 partitions of 1k training points, and Figure 1b uses 20 partitions of 2.5k training points. These partitions allow us to also compute the true bias and variance of the algorithm.[5] To estimate the conditional bias and variance, all models are trained on only one of the 50 (or 20) partitions of the training data but with different random seeds. To estimate the true bias and variance values, each model is trained on a different partition and a different random seed. To calculate the bootstrap estimates of the bias and variance values, one partition out of the total 50 (or 20) partitions is randomly chosen from the training data, then bootstrap estimates are calculated using Algorithm 1.

In both cases, we see that the bootstrapped estimates are more accurate than the conditional samples, but both estimation methods systematically underestimate the variance and overestimate the bias. Additionally, the boostrap bias is more accurate than the bootstrap variance; for this reason, practitioners may prefer to estimate the variance as the total error minus the bootstrapped bias.

---

[4]See Figure 7 in App. C for an empirical analysis of how many partitions are necessary for estimates to converge.

[5]We insist here that these are the bias and variance of the algorithm trained on respectively 1k and 2.5k points, rather than the bias and variance of the algorithm trained on the entire dataset.

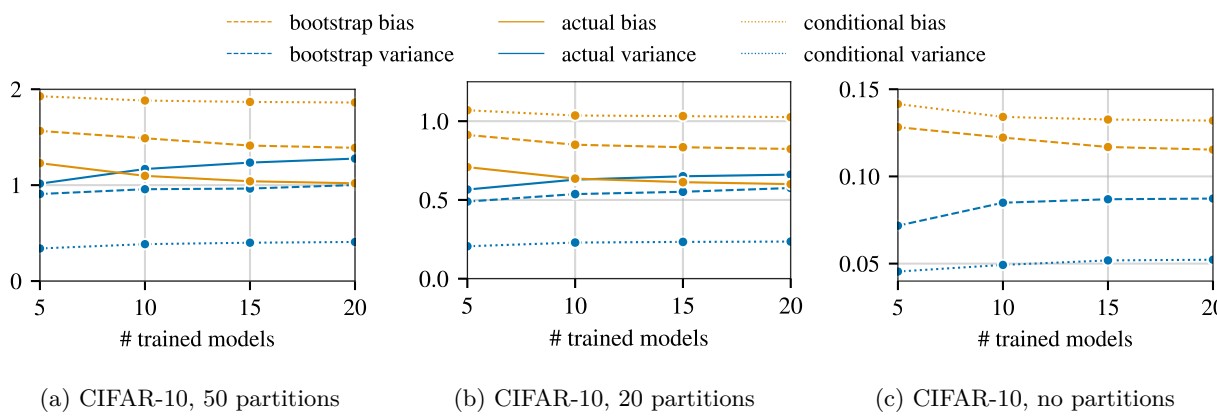

(a) CIFAR-10, 50 partitions     (b) CIFAR-10, 20 partitions     (c) CIFAR-10, no partitions

Figure 1: Conditional and bootstrapped estimates of the bias and variance on variations of the CIFAR-10 dataset. The approximations of the CIFAR-10 dataset in figures (a) and (b) allow us to compute the true bias and variance; in both cases, the bootstrap estimates are more accurate than the conditional estimates. Figure (c) shows the bootstrap and conditional estimates on the true CIFAR-10 dataset, for which the true estimate cannot be computed.

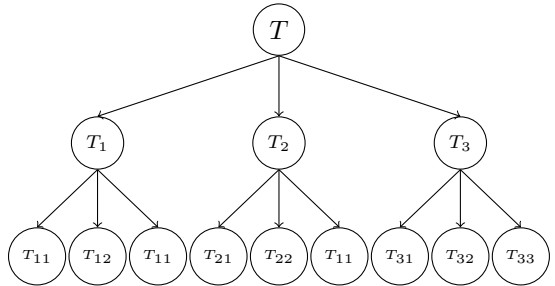

Figure 2: Generating bootstrap samples from a dataset $D$. The child of every node is sampled with replacement from its parent so that all datasets have the same size.

---

**Algorithm 1** Bootstrap estimate of bias (or variance)

---

**Input:** Training set $T$, number of bootstrap samples $B$
**for** $i \in \{1, \ldots, B\}$ **do**
    $T_i \leftarrow \texttt{uniform\_sample}(T)$      ▷ Of size $|T|$
    **for** $j \in \{1, \ldots, B\}$ **do**
        $T_{ij} \leftarrow \texttt{uniform\_sample}(T_i)$   ▷ Of size $|T|$
    $b_i^{(2)} \leftarrow \text{bias}(\{T_{ij}\}_j)$   ▷ Bootstrap estimate for $T_i$
$b^{(1)} \leftarrow \text{bias}(\{T_i\}_i)$   ▷ Bootstrap estimate for $T$
$b^{(2)} \leftarrow \frac{1}{B} \sum_i b_i^{(2)}$
$t \leftarrow b^{(1)}/b^{(2)}$              ▷ Corrective term
$b^{(0)} \leftarrow t b^{(1)}$
**return** Bias estimate $b^{(0)}$.

---

Figure 1c reports bootstrap and conditional bias and variance on the full dataset. Since the gap between actual bias and variance widens as we decrease the number of partitions, it is likely that the true bias dominates the variance in the non-partitioned regime (see also Appendix C).

The rest of our results focus on the conditional bias and variance, for two reasons. First, bootstrap estimates are expensive to compute, requiring training $\mathcal{O}(B^2)$ models. Second, we attempt to match the standard practice of benchmarking on a dataset. Even in cases where a model's loss (rather than bias or variance) is studied, considering performance across *i.i.d.* draws of the training set is not common practice—despite this being the technically correct approach from a frequentist perspective.

# 5 Ensembles and the BV decomposition

We conclude our theoretical analysis by considering ensembles within the context of the bias-variance decomposition for non-symmetric losses. We will distinguish two types of ensembling. *Primal* ensembles average individual model outputs, acting as an empirical approximation of the mean operator. In contrast, *dual* ensembles will approximate the dual mean, by averaging the dual projection of individual model outputs, then casting this average back into primal space.

## 5.1 Primal ensembles

Most often in deep learning, ensembles simply average the outputs of $n$ models that differ in their initialization (Lakshminarayanan et al., 2017). These ensembles are commonly motivated by the desire to reduce the variance of the predictive model. Indeed, for the MSE, we know that ensembles averaging the outputs of models drawn in *i.i.d.* fashion will (a) reduce the variance, and (b) conserve the bias.

We begin by recovering (a) under some additional weak convexity assumptions on the Bregman divergence, but (b) will prove impossible in the general case.

**Proposition 5.1.** *Let $D$ be a Bregman divergence that is* jointly *convex in both variables. Let $X_1, \ldots, X_n$ be $n$ i.i.d. random variables drawn from some unknown distribution, and define $\hat{X} = \frac{1}{n} \sum_i X_i$. Then,*

$$\mathcal{V}\hat{X} = \mathbb{E}D[\mathcal{E}\hat{X}\|\hat{X}] \leq \mathbb{E}D[\mathcal{E}X\|X] = \mathcal{V}X.$$

Both KL divergence and mean squared error are jointly convex, and are special cases of Proposition 5.1.

However, conserving the bias when ensembling requires that the Bregman divergence be symmetric—ensuring that $\mathbb{E}$ and $\mathcal{E}$ are equivalent operators. In the general case, ensembles can either decrease *or* increase the bias.

**Proposition 5.2.** *Let $D$ be the KL divergence. There exists a distribution over predictions $X \in \mathbb{R}^2$ and a label $y \in \{0, 1\}$ such that the bias $D[y\|\mathcal{E}[\cdot]]$ satisfies*

$$D[y\|\mathcal{E}\hat{X}] < D[y\|\mathcal{E}X]$$
$$D[1-y\|\mathcal{E}\hat{X}] > D[1-y\|\mathcal{E}X],$$

*where as above we define the random variable for ensemble predictions $\hat{X} = \frac{1}{n} \sum_i X_i$, and by abuse of notation we conflate $y \in \{0, 1\}$ with its one-hot representation.*

*Remark* 5.1. Despite Proposition 5.2, ensembles reduce the overall cross-entropy loss due to Jensen's inequality.

That standard ensembles do not preserve the bias (and can, in fact, increase it) is a strong departure from what one might naively expect. Thus, it is natural to seek an ensemble method that would maintain the following two behaviors: the variance decreases, and the bias is conserved.

## 5.2 Dual ensembles

To keep the bias $D[\mathbb{E}Y\|\mathcal{E}X]$ unchanged under ensembling, it is clearly sufficient to ensemble in such a way that the ensemble predictor $\hat{X}$ satisfies the equality $\mathcal{E}\hat{X} = \mathcal{E}X$. This is also sufficient to reduce the variance.

**Proposition 5.3.** *Let $D$ be any Bregman divergence. Let $X_1, \ldots, X_n$ be $n$ i.i.d. random variables drawn from some unknown distribution, and define the dual ensemble*

$$\hat{X} = \left(\frac{1}{n} \sum_i X_i^*\right)^*.$$

*This operation ensures that $\mathcal{E}X = \mathcal{E}\hat{X}$. Furthermore, dual ensembling reduces the variance and conserves the bias: for any independent label variable $Y$ over $\mathcal{X}$, we have*

$$D[\mathbb{E}Y \| \mathcal{E}\hat{X}] = D[\mathbb{E}Y \| \mathcal{E}X]$$
$$\mathbb{E}D[\mathcal{E}\hat{X} \| \hat{X}] \leq \mathbb{E}D[\mathcal{E}X \| X].$$

Compared to Proposition 5.1 for primal ensembles, dual ensembles do not require that $D$ be jointly convex to guarantee variance reduction; the natural convexity of $D$ in its first argument is sufficient.

*Remark* 5.2. Under the KL divergence, dual ensembles amount to averaging model logits instead of model probabilities (sometimes referred to as *geometric* averaging).

$$\hat{X} = \text{softmax}\left(\frac{1}{n} \sum_i \log(X_i)\right).$$

Additional properties of dual ensembling under the KL divergence have been discussed in (Brofos & Shu, 2019); that it preserves bias has been mentioned briefly in (Dietterich, 2005).

*Remark* 5.3. Reordering Equation (3) for the KL divergence, we can write the loss of an ensemble of models averaged in logit space as the difference between the average individual model loss and the variance, thus recovering the ensemble diversity regularizer from (Webb et al., 2020). Consider a noiseless label $Y = y$ for the cross-entropy loss, and let the random variable $X$ be uniformly distributed over $n$ different models $x_1, \ldots, x_n$. Writing $\hat{x} := \mathrm{softmax}(\frac{1}{n} \sum_i \log x_i)$ the (dual) ensemble prediction, Equation (3) then yields

$$\frac{1}{n} \sum_i \mathrm{CE}[y\|x_i] = \mathrm{CE}[y, \hat{x}] + \frac{1}{n} \sum_i \mathrm{CE}[\hat{x}\|x_i].$$

Reordering, we can write the ensemble loss $\mathrm{CE}[y, \hat{x}]$ as the difference between the average individual model loss and the regularizer in (Webb et al., 2020, Eq. (5)) with regularization strength $\lambda = 1$:

$$\mathrm{CE}[y, \hat{x}] = \frac{1}{n} \sum_i \mathrm{CE}[y\|x_i] - \frac{1}{n} \sum_i \mathrm{CE}[\hat{x}\|x_i].$$

# 6 Empirical analysis of ensembles under the cross-entropy loss

Choosing the Bregman divergence $D$ to be the KL divergence, we know that both primal and dual ensembles reduce the total loss and variance. Furthermore, dual ensembles do not affect the bias, but primal ensembles can either increase or decrease the bias. We now analyze how primal and dual ensembles compare in practice when ensembling neural networks.

## 6.1 Comparing primal and dual ensembles

We begin by an empirical validation of the expected behaviors of primal and dual ensembles. Figures 3a and 3b show the evolution of the total loss, bias, and variance of ensembles of independent WRNs 28–10 under the cross-entropy loss on the associated Cifar test sets; Figure 3c stratifies the decomposition by corruption intensity on the corrupted Cifar-100 (Hendrycks & Dietterich, 2019) test set.

For both primal and dual ensembles, the bias dominates the variance. As these are conditional estimates subject to the estimation error described in Proposition 4.1, we cannot affirm that the true bias dominates the true variance, although this is plausible based on the conclusions of Figure 1.

As expected, the dual bias is independent of ensemble size. The variance is reduced both by primal and dual ensembling, and is slightly lower for dual ensembles. Finally, bias and variance gaps between primal and dual ensembles widen as the corruption severity increases (Figure 3c). Surprisingly, the primal bias *decreases* when ensembling; Proposition 5.2 states that primal ensembling can affect the bias, but not necessarily by reducing it.[6] However, since primal ensembling reduces bias on these datasets, and because the gap between primal and dual variances is small, the primal NLL is lower; this behavior has been noted previously in (Brofos & Shu, 2019). To understand this, we turn to a pointwise analysis.

Consider $n$ models that assign probabilities $p_1, \ldots, p_n$ to the true class. In the two-class setting, we can exactly write out the primal NLL as $-\log \frac{p_1 + \ldots + p_n}{n}$, and the dual NLL as $-\log \frac{(p_1 \cdots p_n)^{1/n}}{(p_1 \cdots p_n)^{1/n} + ((1-p_1) \cdots (1-p_n))^{1/n}}$.

Fixing $p_1, \ldots, p_{n-1} > 0$, and considering $f(p_n) = primal\_loss(p_1, \ldots, p_n) - dual\_loss(p_1, \ldots, p_n)$, we have

$$-\infty \leq f(p_n) \leq -\log \frac{p_1 + \ldots + p_{n-1}}{n}. \tag{4}$$

The lower bound of Equation (4) is achieved for $p_n \to 0$: a single model in the ensemble can make the dual loss arbitrarily worse than the primal loss. In contrast, the same single model can at best provide a bounded

---

[6]Note that the terms shown in Figure 3 are averaged over the entire test set. On individual points, we will see below that the bias can be increased *or* decreased by ensembling.

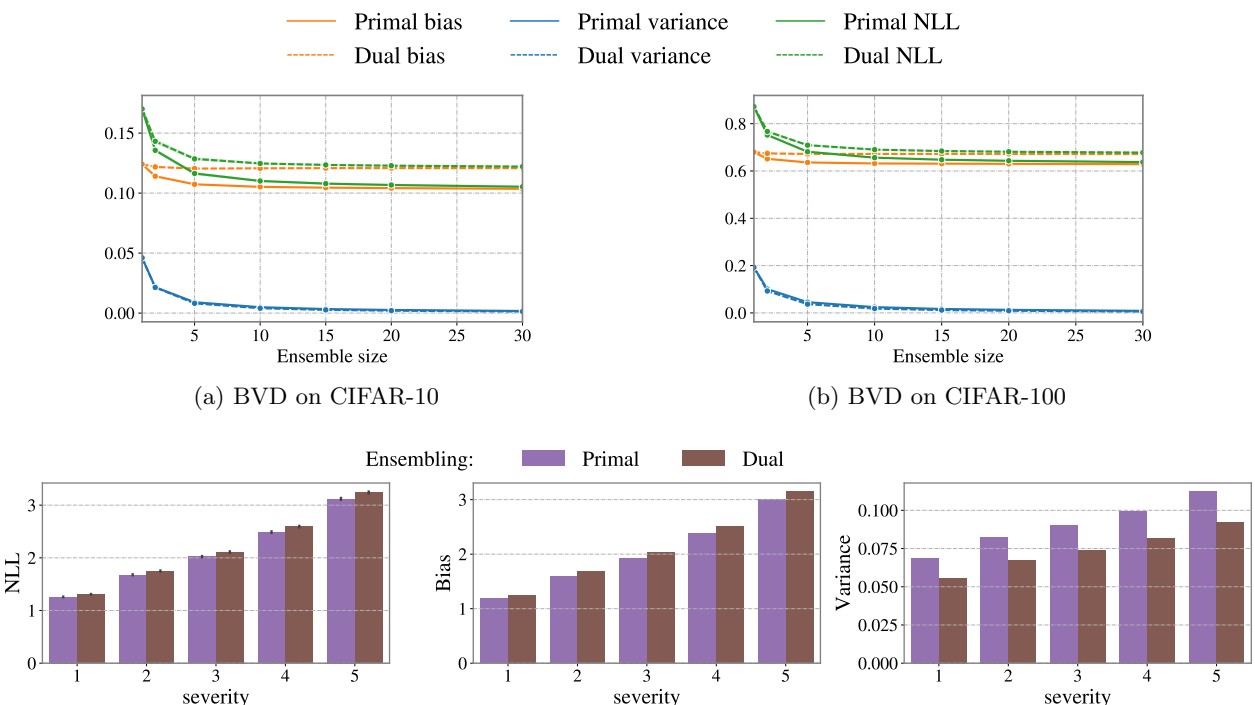

(a) BVD on CIFAR-10

(b) BVD on CIFAR-100

(c) BVD on Cifar100 corrupted datasets, stratified by corruption severity, for an ensemble size of 5.

Figure 3: Conditional bias, variance, and NLL of ensembles of WRNs 28–10 (estimated using 20 draws of each ensemble size). The dual bias remains constant as a function of the ensemble size, while the primal bias is reduced. The variance is reduced slightly more for dual ensembles, but at an overall magnitude much smaller than the bias.

improvement for dual ensembles over primal ensembles. Thus, even a single point in a dataset with a large dual loss is theoretically sufficient to make primal ensembling outperform dual ensembling on average.

After some simple arithmetic manipulations, we obtain the following equivalent condition under which primal ensembling achieves a lower NLL than dual ensembling in the two-class setting:

$$\frac{\sum p_i}{\sum (1-p_i)} \geq \frac{\prod p_i^{1/n}}{\prod (1-p_i)^{1/n}}. \tag{5}$$

Inequality (5) of arithmetic and geometric mean ratios can be further simplified for ensembles of two models.

**Proposition 6.1.** *Let $p_1$ and $p_2$ be the probabilities assigned by two models to the true class $y$. Primal ensembling these models will achieve a lower cross-entropy loss than dual ensembling if and only if $p_2 \leq 1 - p_1$.*

In other words, primal ensembles are favorable when there is significant disagreement between ensemble members. This is not surprising: for primal ensembles, all predictions contribute similarly to the ensemble prediction, whereas dual ensembles are more sensitive to extreme predictions ($p \to 0$ or $p \to 1$), and in particular to extreme incorrect predictions ($p \to 0$)—see also Figure 16 (Appendix H).

We confirm this in Figure 4, visualizing the pointwise BVD for ensembles of size $n = 5$ on CIFAR-100. As expected, we see that dual ensembles are more likely to provide extreme probability estimates than primal ensembles (Figure 4b). When primal ensembles achieve a lower NLL than dual ensembles (Figure 4c), both primal and dual ensembles assign probabilities close to zero to the true class; however, dual probabilities are more concentrated around zero. Conversely, when dual ensembles achieve the lower loss, primal and dual probabilities are more spread out, and dual probabilities tend closer to one (Figure 4d). Finally, for the ensembles analyzed in Figure 4, the dual NLL is lower than the primal NLL on almost 85% of datapoints; on such points, we have $|\text{NLL}_{primal} - \text{NLL}_{dual}| \approx 0.05$, whereas on the 15% of datapoints where dual NLL is higher than the primal NLL, we have $|\text{NLL}_{primal} - \text{NLL}_{dual}| \approx 0.42$. This is in line with our earlier analysis,

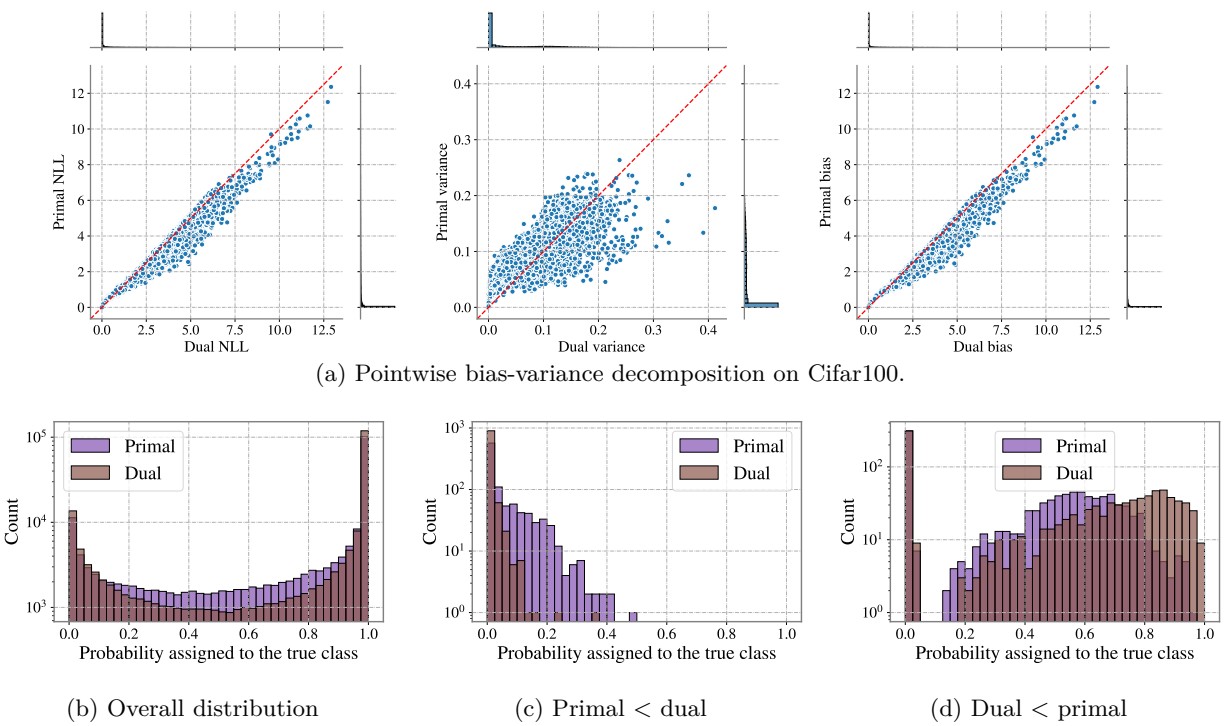

(a) Pointwise bias-variance decomposition on Cifar100.

(b) Overall distribution        (c) Primal < dual        (d) Dual < primal

Figure 4: Pointwise analysis of the bias-variance decomposition on the CIFAR-100 test set, using 20 draws of size-5 ensembles. (a) Decomposition over the full test set. Primal quantities (NLL, bias and variance) tend to be lower than their dual counterparts, except for when the loss is very close to zero, in which case dual quantities are smaller. (b) Empirical distribution of primal and dual probabilities assigned to the correct class over all CIFAR-100 test points; extreme predictions (close to 0 or 1) are more common for dual ensembles. (b) Empirical distribution of probabilities assigned to the correct class for the 50 points where the signed difference between primal and dual loss is smallest; (c) for the 50 points where the signed difference between primal and dual loss is largest.

which showed that the asymmetry between primal and dual NLL allows for only a few points with much worse dual NLL to influence the overall average performance.

We conclude this section by noting that Equation (5) still requires knowledge of the true labels. We leave to future work the question of whether one can predict whether primal or dual ensembling will achieve a lower loss *without* access to the true labels.

## 6.2 Primal ensembles of different model classes

We conclude our analysis by investigating the behavior of (primal) ensembles that average over network width or network depth as well as random seeds; this has been shown to improve upon ensembling only over the random seed (Zaidi et al., 2021). Based on our analysis, we seek to evaluate whether this improvement over standard ensembles is due to faster bias reduction. (Yang et al., 2020) already showed that ensembles of larger models have smaller bias; here, we are interested in whether the bias reduction as a function of ensemble size is more *efficient* when ensembling over different architectures.

We begin with ensembles over different depths, training 100 WRNs of depths of 28, 40, and 52 each, following the prescribed pattern $depth = 6d + 4$; the width multiplier is set to 10. Figure 5 shows indeed that, for ensembles of different depths, the bias decreases faster as a function of the ensemble size compared to ensembles of fixed depth and equivalent model size. Ensembles of different depths unsurprisingly have higher variance; however, this increase in variance is less than the decrease in bias, as shown by the overall NLL. The bias, variance, and NLL values of size-1 ensembles is provided in Appendices E and F.

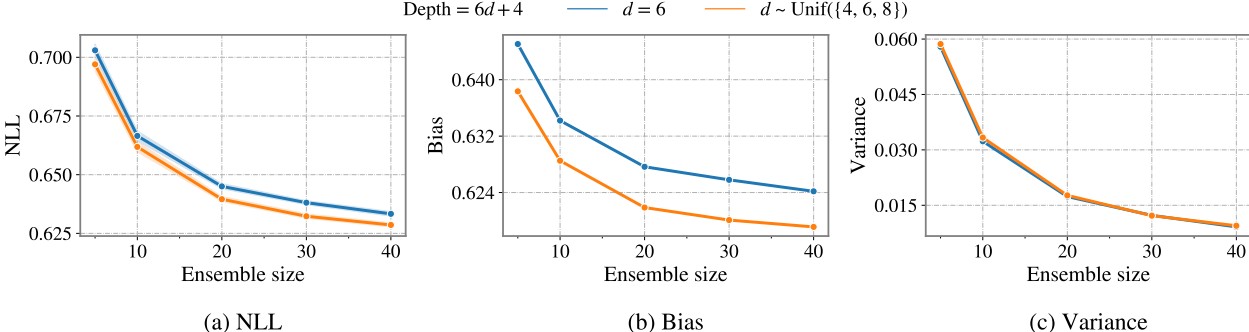

Figure 5: Bias, variance, and NLL on CIFAR-100 (estimated using 20 ensemble draws). Networks are trained with different random seeds and different depths and then averaged in probability space (*primal ensembles*). Averaging over depths dramatically reduces the bias, but in turn increases the variance. However, the increase in variance is much smaller than the decrease in bias, and ensembles over multiple depths outperform ensembles of fixed depth.

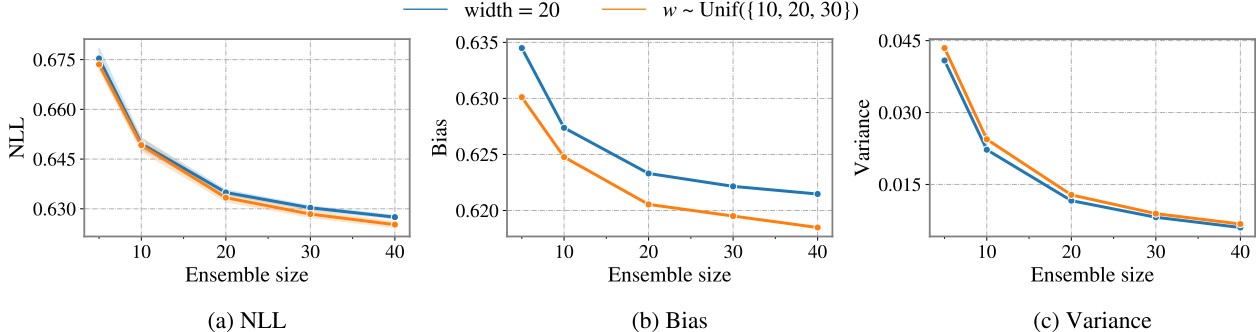

Figure 6: Bias, variance and NLL on CIFAR-100 (estimated using 20 ensemble draws). Networks are trained with different initial random seeds and different widths and then averaged in probability space (*primal ensembles*). Averaging over the width improves the bias at the cost of the variance.

A similar conclusion holds for ensembles of different widths (Figure 5), where this time we fix the depth to $6d + 4 = 28$ and let the width multiplier take values in $\{10, 20, 30\}$. Appendices E and F include results for CIFAR-10 and corrupted datasets, with similar conclusions—only on the CIFAR-100 corrupted data is a faster decrease in bias unclear.

## 7 Conclusion

Ensembles of deep classifiers achieve state-of-the-art performance across a variety of benchmark tasks. Where ensembles have previously been analyzed for regression models through the lens of the bias-variance decomposition, applying this decomposition proves more complicated for non-symmetric losses—such as the KL divergence-based losses used in the vast majority of classification tasks. Pfau (2013) generalized the BVD to Bregman divergences (including the KL divergence). However, whereas the "central" prediction for the MSE decomposition is simply the expected prediction $\mathbb{E}X$, the corresponding quantity for arbitrary Bregman divergences is defined as the minimizer of an expected divergence, a term much less amenable to analysis.

We begin by resolving this difficulty by viewing the Bregman decomposition through the lens of convex conjugates, showing that Bregman divergences implicitly defines primal and dual spaces, distinct when the divergence is asymmetric. This perspective allows us to redefine the central prediction simply as the primal projection of the expectation over dual predictions.

Using this reparameterization, we easily show that the model variance resulting from the bias-variance decomposition satisfies a generalized law of total variance. We also exactly quantify the error term in estimating the bias and variance terms when such estimates are implicitly conditioned on an uncontrolled source of randomness (oftentimes, the randomness in the training data).

The existence of primal and dual spaces for the bias-variance decomposition also has key implications when building ensembles. Ensembling in primal space corresponds to simply averaging model predictions; this is the usual setting for ensembles of neural network classifiers. We show that primal ensembles deviates from standard assumptions: although they will reduce the variance under gentle assumptions, primal ensembling may either increase or decrease the bias. Conversely, ensembling in dual space recovers expected behaviors, always reducing the variance and leaving the bias unchanged.

Comparing different estimates of the bias and variance, we show that conditional estimates incur an irreducible error, and that empirically the conditional bias dominates the variance for WRNs on Cifar. Bootstrapped and partitioned estimates suggest that this holds true for the true bias and variance on the full test set.

Finally, we use the bias-variance decomposition to investigate the empirical behavior of neural network ensembles under the cross-entropy loss. Experimentally, we observe that primal ensembling (a) *reduces* the average bias, and (b) achieves a variance reduction of similar magnitude to dual ensembles, thus achieving a lower overall NLL. To understand this behavior, we compare primal and dual ensembles on individual inputs, showing that dual ensembles are more sensitive to extreme predictions, and incorrect predictions particularly. Finally, we turn to (primal) ensembles of neural networks over different architectures, hypothesizing that their improved performance may be in part explained by the behavior of the bias. Empirically, we confirm that ensembling over architectures provides a faster bias reduction than ensembling only over random seeds.

Our analysis opens several avenues for future work. Of particular interest is identifying whether the average bias reduction (and corresponding lower loss) of primal ensembles under the cross-entropy loss is common to other datasets and tasks, as well as applying our analysis to learned Bregman divergences (Cilingir et al., 2020; Siahkamari et al., 2020; Lu et al., 2022). Analyzing other common ensembling methods used in deep learning and understanding their affect on the bias and variance is also of independent interest. Finally, we found that primal ensembling achieves much greater variance reduction than bias reduction, despite bias being the dominating term—raising the question of whether the insights developed in this work can help derive ensembling methods with greater bias reduction.

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

# A   Proofs

**Proposition A.1** (Generalized triangle inequality for Bregman divergences)**.** *For any $x, y, z$ in the domain of $F$, we have $D(x, z) = D(x, y) + D(y, z) + \langle \nabla F(y) - F(z), x - y \rangle$.*

**Lemma 3.2.** *Let $X, Z$ be random variables over $\mathcal{X}$, and define the conditional variance $\mathcal{V}[X|Z] := \mathbb{E}\big[D[\mathcal{E}(X|Z) \parallel X]\big|Z\big]$. The variance $\mathcal{V}X := \mathbb{E}D[\mathcal{E}X\|X]$ satisfies a generalized law of total variance:*

$$\mathcal{V}[X] = \mathbb{E}[\mathcal{V}[X|Z]] + \mathcal{V}[\mathcal{E}[X|Z]].$$

*Proof.* The proof follows from the generalized triangle inequality for Bregman divergences (Proposition A.1).

$$
\begin{aligned}
\mathcal{V}X &= \mathbb{E}D[\mathcal{E}X\|X] \\
&= \mathbb{E}_Z\mathbb{E}\Big[D[\mathcal{E}X\|X]\Big|Z\Big] \\
&= \mathbb{E}_Z\mathbb{E}\Big[D[\mathcal{E}X\|\mathcal{E}(X|Z)] + D[\mathcal{E}(X|Z)\|X] + \langle \nabla F(\mathcal{E}(X|Z)) - \nabla F(X), \mathcal{E}X - \mathcal{E}(X|Z)\rangle\Big|Z\Big] \\
&= \mathbb{E}_Z\mathbb{E}\Big[D[\mathcal{E}X\|\mathcal{E}(X|Z)] + D[\mathcal{E}(X|Z)\|X] + \langle (\mathcal{E}(X|Z))^* - X^*, \mathcal{E}X - \mathcal{E}(X|Z)\rangle\Big|Z\Big] \\
&= \mathbb{E}_Z\mathbb{E}\Big[D[\mathcal{E}X\|\mathcal{E}(X|Z)] + D[\mathcal{E}(X|Z)\|X] + \langle \mathbb{E}[X^*|Z] - X^*, \mathcal{E}X - \mathcal{E}(X|Z)\rangle\Big|Z\Big] \\
&= \mathbb{E}_Z D[\underbrace{\mathcal{E}X}_{\mathcal{E}[\mathcal{E}(X|Z)]}\|\mathcal{E}(X|Z)] + \mathbb{E}_Z\underbrace{\Big[\mathbb{E}D[\mathcal{E}(X|Z)\|X]\Big|Z\Big]}_{\mathcal{V}(X|Z)} + \underbrace{\mathbb{E}_Z\langle \mathbb{E}[X^*|Z] - \mathbb{E}[X^*|Z], \mathcal{E}X - \mathcal{E}(X|Z)\rangle}_{=0} \\
&= \mathcal{V}[\mathcal{E}(X|Z)] + \mathbb{E}[\mathcal{V}(X|Z)].
\end{aligned}
$$

$\square$

**Proposition 4.1.** *Let $X, Z$ be two random variables over $\mathcal{X}$; to simplify notation, we assume that the label $Y$ is deterministic ($Y = y \in \mathcal{X}$). Applying Equation (3) to $X|Z$ then taking the expectations over $Z$ yields an alternate BVD:*

$$\mathbb{E}D[y\|X] = \underbrace{\mathbb{E}_Z D[y\|\mathcal{E}(X|Z)]}_{\text{Conditional bias: } \mathrm{Bias}_Z} + \underbrace{\mathbb{E}_Z\mathbb{E}\Big[D[\mathcal{E}(X|Z)\|X]\Big|Z\Big]}_{\text{Conditional variance: } \mathrm{Var}_Z}.$$

*The conditional bias (resp. variance) overestimates (resp. underestimates) their respective total values by the fixed quantity $\mathbb{E}_Z D[\mathcal{E}X\|\mathcal{E}(X|Z)]$:*

$$\mathrm{Bias}_Z = \text{total bias} + \mathbb{E}_Z D[\mathcal{E}X\|\mathcal{E}(X|Z)] \qquad \mathrm{Var}_Z = \text{total variance} - \mathbb{E}_Z D[\mathcal{E}X\|\mathcal{E}(X|Z)].$$

*Proof.* Applying Equation (3) to the conditional bias $\mathbb{E}_Z D[y\|\mathcal{E}(X|Z)]$, we have

$$
\begin{aligned}
\mathbb{E}_Z D[y \parallel \mathcal{E}(X \mid Z)] &= D\Big[y \parallel \mathcal{E}[\mathcal{E}(X|Z)]\Big] + \mathbb{E}_Z D\Big[\mathcal{E}[\mathcal{E}(X|Z)] \parallel \mathcal{E}(X|Z)\Big] \\
&= D[y \parallel \mathcal{E}X] + \mathbb{E}_Z D[\mathcal{E}X \parallel \mathcal{E}(X|Z)],
\end{aligned}
$$

where the last equality stems from the law of iterated expectations for $\mathcal{E}$. The result for the variance terms follows immediately, as conditional bias and variance have the same sum as the full bias and variance. $\square$

**Proposition 5.1.** *Let $D$ be a Bregman divergence that is* jointly *convex in both variables. Let $X_1, \ldots, X_n$ be $n$ i.i.d. random variables drawn from some unknown distribution, and define $\hat{X} = \frac{1}{n}\sum_i X_i$. Then,*

$$\mathcal{V}\hat{X} = \mathbb{E}D[\mathcal{E}\hat{X}\|\hat{X}] \le \mathbb{E}D[\mathcal{E}X\|X] = \mathcal{V}X.$$

*Proof.* Let $D : \mathcal{X} \times \mathcal{X} \to \mathbb{R}^+$ be a Bregman divergence jointly convex in both variables. Let $\hat{X} = \frac{1}{n} \sum_i X_i$, where the $X_i$ are *i.i.d.*. By convexity, for any $z \in \mathcal{X}$,

$$D[z \parallel \hat{X}] \leq \frac{1}{n} \sum_i D[z \parallel X_i]$$

$$\mathbb{E}D[z \parallel \hat{X}] \leq \frac{1}{n} \sum_i \mathbb{E}D[z \parallel X_i] = \mathbb{E}D[z \parallel X]$$

$$\min_z \mathbb{E}D[z \parallel \hat{X}] \leq \min_z \mathbb{E}D[z \parallel X].$$

As $\mathcal{E}X = \arg\min_z \mathbb{E}D[z \parallel X]$, $\min_z \mathbb{E}D[z \parallel X] = \mathbb{E}D[\mathcal{E}X \parallel X]$, concluding the proof. $\square$

**Proposition 5.2.** *Let $D$ be the KL divergence. There exists a distribution over predictions $X \in \mathbb{R}^2$ and a label $y \in \{0, 1\}$ such that the bias $D[y \| \mathcal{E}[\cdot]]$ satisfies*

$$D[y\|\mathcal{E}\hat{X}] < D[y\|\mathcal{E}X]$$

$$D[1 - y\|\mathcal{E}\hat{X}] > D[1 - y\|\mathcal{E}X],$$

*where as above we define the random variable for ensemble predictions $\hat{X} = \frac{1}{n} \sum_i X_i$, and by abuse of notation we conflate $y \in \{0, 1\}$ with its one-hot representation.*

*Proof.* For any one-hot label $y \in \{0, 1\}$ and probability vector $x$, we have $\mathrm{KL}[y\|x] = \log x_y$, and $\mathrm{KL}[1 - y\|x] = \log(1 - x_y)$. As $x \to \log 1 - x$ is decreasing, it suffices to prove that there exists a distribution $\mathcal{D}$ such that $\mathrm{KL}[y\|\mathcal{E}\hat{X}] \neq \mathrm{KL}[y\|\mathcal{E}X]$. In fact, it suffices to prove the existence of a distribution $\mathcal{D}$ such that $\mathcal{E}X \neq \mathcal{E}\hat{X}$.

For the cross-entropy loss, we know[7] that $\mathcal{E}X = \mathrm{softmax}(\mathbb{E}\log X)$. Let $\mathcal{D}$ be the distribution that assigns equal probability to $x = (0.8, 0.2)$ and $x = (0.6, 0.4)$, and is zero elsewhere. The equivalent ensemble distribution assigns $1/4$ probability to $(0.8, 0.2)$ and $(0.6, 0.4)$, and $1/2$ probability to $(0.7, 0.3)$. A simple numerical computation then shows that $\mathcal{E}X \neq \mathcal{E}\hat{X}$, concluding our proof. $\square$

**Proposition 5.3.** *Let $D$ be any Bregman divergence. Let $X_1, \ldots, X_n$ be $n$ i.i.d. random variables drawn from some unknown distribution, and define the dual ensemble*

$$\hat{X} = \left( \frac{1}{n} \sum_i X_i^* \right)^*.$$

*This operation ensures that $\mathcal{E}X = \mathcal{E}\hat{X}$. Furthermore, dual ensembling reduces the variance and conserves the bias: for any independent label variable $Y$ over $\mathcal{X}$, we have*

$$D[\mathbb{E}Y \parallel \mathcal{E}\hat{X}] = D[\mathbb{E}Y \parallel \mathcal{E}X]$$

$$\mathbb{E}D[\mathcal{E}\hat{X} \parallel \hat{X}] \leq \mathbb{E}D[\mathcal{E}X \parallel X].$$

*Proof.* To preserve bias, it suffices to have $\mathcal{E}\hat{X} = \mathcal{E}X$. By definition of $\hat{X}$, we have

$$\mathcal{E}\hat{X} = \left( \mathbb{E}\hat{X}^* \right)^* = \left( \mathbb{E}\left[ \frac{1}{n} \sum_i X_i^* \right] \right)^* = (\mathbb{E}X^*)^* = \mathcal{E}X. \tag{6}$$

---

[7]See, *e.g.*, (Yang et al., 2020).

We now focus on the variance. Successively,

$$
\begin{aligned}
\mathbb{E} D_F[\mathcal{E}\hat{X}\|\hat{X}] &= \mathbb{E} D_F[\mathcal{E}X\|\hat{X}] && \text{(by } \mathcal{E}X = \mathcal{E}\hat{X}) \\
&= \mathbb{E} D_{F^*}[\hat{X}^*\|(\mathcal{E}X)^*] && \text{(by Proposition 3.1)} \\
&= \mathbb{E} D_{F^*}\Big[\frac{1}{n}\sum_i X_i^*\|(\mathcal{E}X)^*\Big] \\
&\overset{(a)}{\leq} \frac{1}{n}\sum_i \mathbb{E} D_{F^*}\Big[X_i^*\|(\mathcal{E}X)^*\Big] && \text{(by convexity of } D_{F^*} \text{ in its first argument)} \\
&\leq \frac{1}{n}\sum_i \mathbb{E} D_F\Big[\mathcal{E}X\|X_i\Big] \\
&\leq D_F[\mathcal{E}X\|X].
\end{aligned}
$$

$\square$

**Proposition 6.1.** *Let $p_1$ and $p_2$ be the probabilities assigned by two models to the true class $y$. Primal ensembling these models will achieve a lower cross-entropy loss than dual ensembling if and only if $p_2 \leq 1-p_1$.*

*Proof.* When $n = 2$, inequality (5) holds if and only if

$$
(1-p_1)(1-p_2)(p_1+p_2)^2 - p_1 p_2 (2 - (p_1 + p_2))^2 \geq 0
$$
$$
\Longleftrightarrow \qquad (p_1 - p_2)^2 (1 - (p_1 + p_2)) \geq 0.
$$

Thus, the primal loss is smaller than the dual loss if and only if $p_1 = p_2$ or $p_2 \leq 1 - p_1$. $\square$

# B Experimental Details

The models used in this work are wide residual networks (WRN-28-10) (Zagoruyko & Komodakis, 2016) with the cross-entropy loss unless specified otherwise. We train models with SGD + momentum to optimize the cross-entropy loss. We use the learning rate schedule, batch size, and data augmentations specified in the deterministic baseline provided by Nado et al. (2021).

## C   Partitioned estimates of the bias and variance

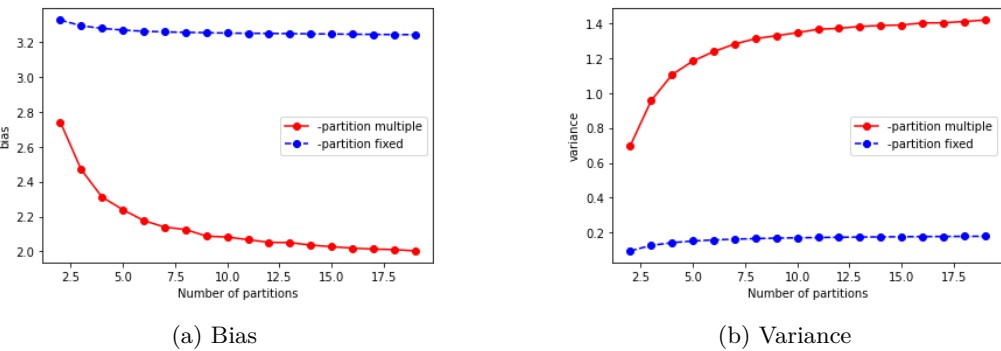

(a) Bias

(b) Variance

Figure 7: Bias and variance of a smaller WRN-16-5 over the CIFAR-100 dataset. We create 20 partitions of the CIFAR-100 dataset, and estimate the bias either by conditioning on a partition (partition fixed/conditional estimate), or by including the partition into the expectations that define the bias and variance (partition multiple; converges to the true estimate). We see that it takes $\geq 10$ partitions for the estimates of bias and variance to begin converging, and that the converged values appear to still show the bias dominating the variance.

## D   Bias-variance decomposition on SVHN and corrupted Cifar

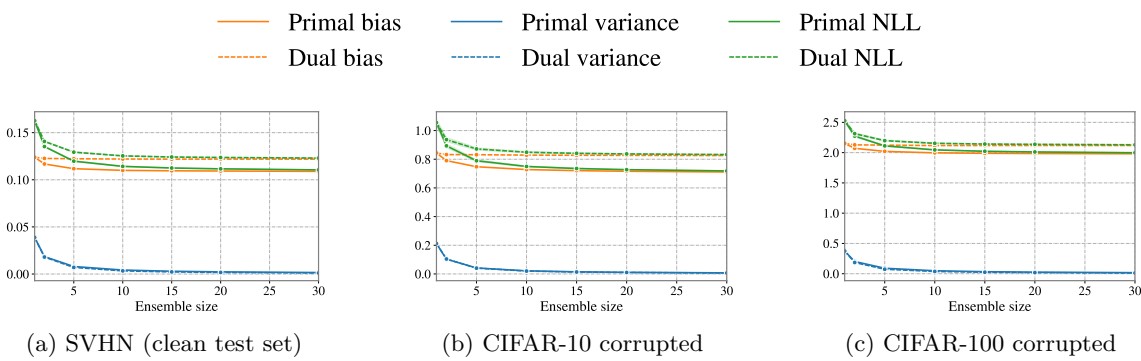

(a) SVHN (clean test set)

(b) CIFAR-10 corrupted

(c) CIFAR-100 corrupted

Figure 8: Conditional bias, variance, and NLL of primal and dual WRN-28–10 ensembles on the SVHN test set (left) and the corrupted Cifar10 and Cifar100 test sets (averaged over all corruptions). The comparison between primal and dual ensembling methods remains similar, although are values are increased due to the dataset shift.

# E    Additional depth experiments

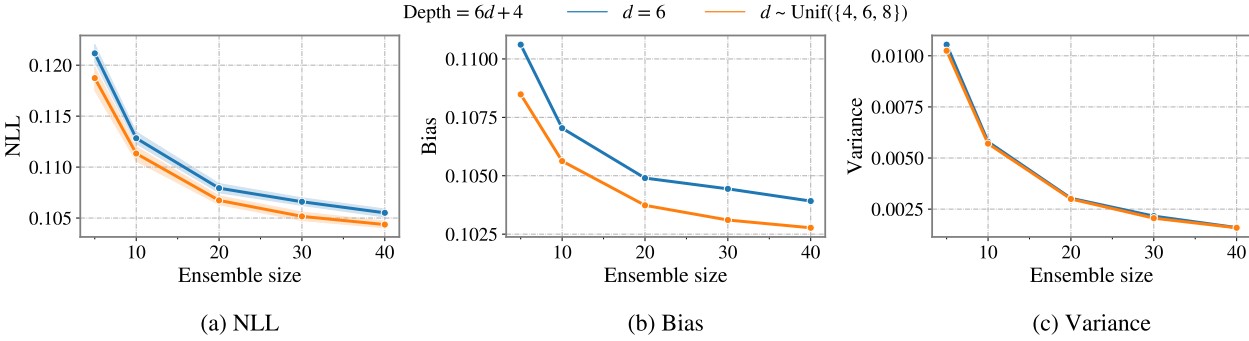

Figure 9: (Primal) ensembling over depths: bias-variance decomposition on the Cifar10 clean test set.

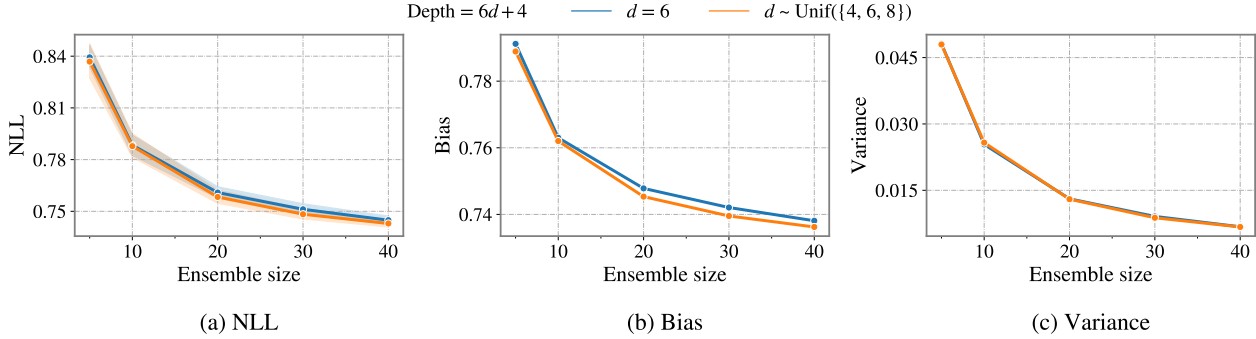

Figure 10: (Primal) ensembling over depths: bias-variance decomposition on the Cifar10 corrupted test set.

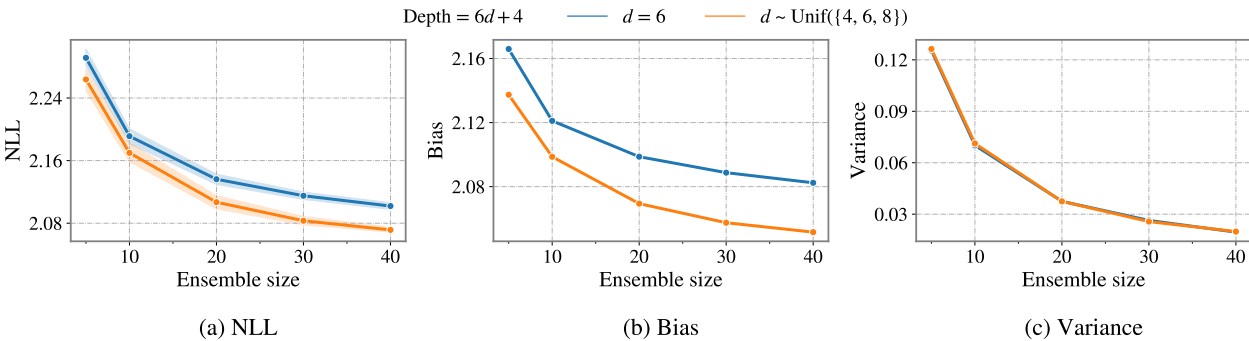

Figure 11: (Primal) ensembling over depths: bias-variance decomposition on the Cifar100 corrupted test set.

| Depth | NLL | Bias | Variance |
|---|---|---|---|
| 6 | 0.181 | 0.127 | 0.053 |
| Uniform | 0.177 | 0.125 | 0.051 |

(a) Clean test set

| Depth | NLL | Bias | Variance |
|---|---|---|---|
| 6 | 1.170 | 0.923 | 0.247 |
| Uniform | 1.120 | 0.885 | 0.235 |

(b) Corrupted test set

Table 1: Bias, variance, and NLL values for size-1 ensembles for varying WRN depths on Cifar10

| Depth | NLL | Bias | Variance |
|---|---|---|---|
| 6 | 0.971 | 0.717 | 0.254 |
| Uniform | 0.980 | 0.718 | 0.262 |

(a) Clean test set

| Depth | NLL | Bias | Variance |
|---|---|---|---|
| 6 | 2.887 | 2.355 | 0.532 |
| Uniform | 2.917 | 2.372 | 0.547 |

(b) Corrupted test set

Table 2: Bias, variance, and NLL values for size-1 ensembles for varying WRN depths on Cifar100

## F    Additional width experiments

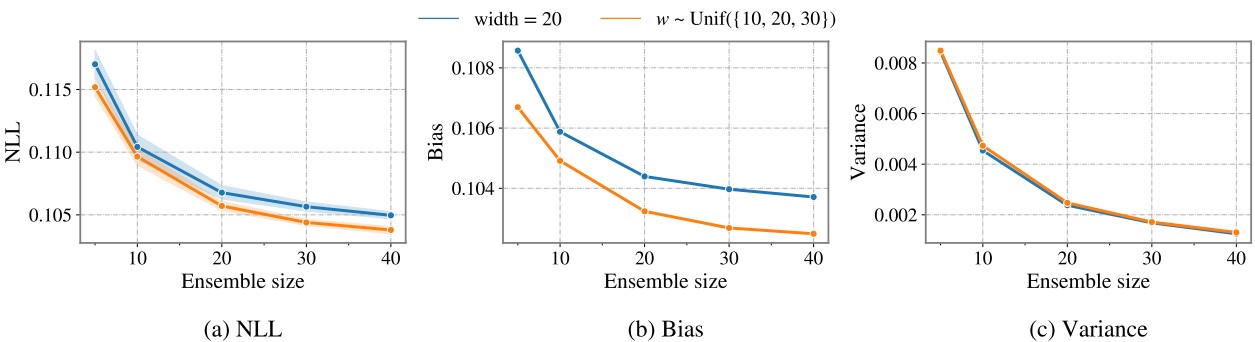

(a) NLL    (b) Bias    (c) Variance

Figure 12: (Primal) ensembling over widths: bias-variance decomposition on the Cifar10 clean test set.

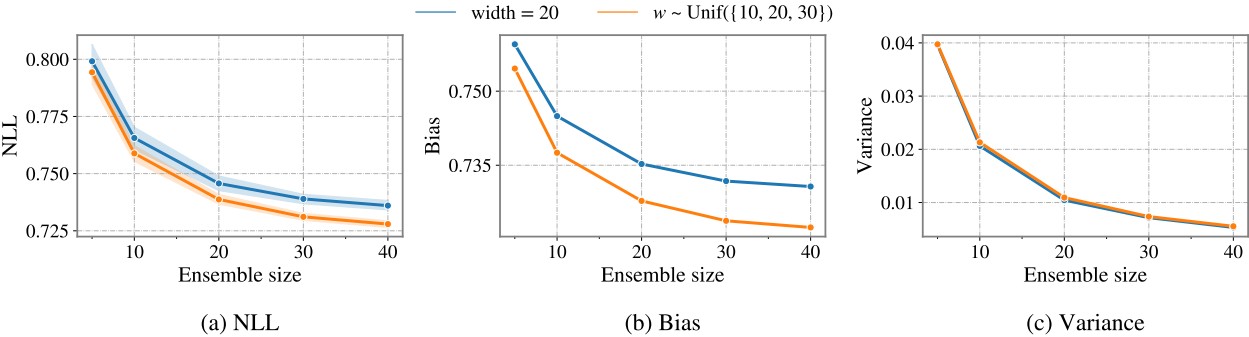

(a) NLL    (b) Bias    (c) Variance

Figure 13: (Primal) ensembling over widths: bias-variance decomposition on the Cifar10 corrupted test set.

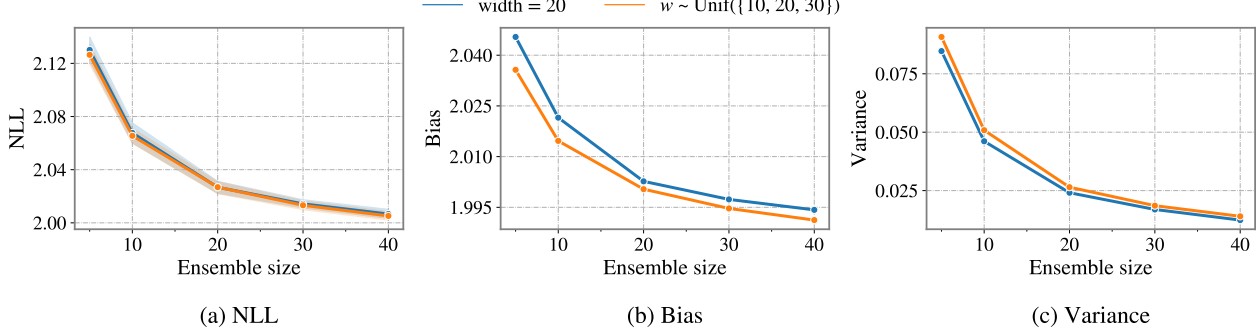

(a) NLL    (b) Bias    (c) Variance

Figure 14: (Primal) ensembling over widths: bias-variance decomposition on the Cifar100 corrupted test set.

| Width | NLL | Bias | Variance |
|---|---|---|---|
| 20 | 0.164 | 0.123 | 0.041 |
| Uniform | 0.164 | 0.121 | 0.043 |

(a) Clean test set

| Width | NLL | Bias | Variance |
|---|---|---|---|
| 20 | 1.044 | 0.848 | 0.196 |
| Uniform | 1.040 | 0.841 | 0.199 |

(b) Corrupted test set

Table 3: Bias, variance, and NLL values for size-1 ensembles for varying WRN widths on Cifar10

| Width | NLL | Bias | Variance |
|---|---|---|---|
| 20 | 0.864 | 0.683 | 0.180 |
| Uniform | 0.877 | 0.684 | 0.193 |

(a) Clean test set

| Width | NLL | Bias | Variance |
|---|---|---|---|
| 20 | 2.529 | 2.166 | 0.364 |
| Uniform | 2.574 | 2.176 | 0.399 |

(b) Corrupted test set

Table 4: Bias, variance, and NLL values for size-1 ensembles for varying WRN widths on Cifar100

# G    Expected calibration error

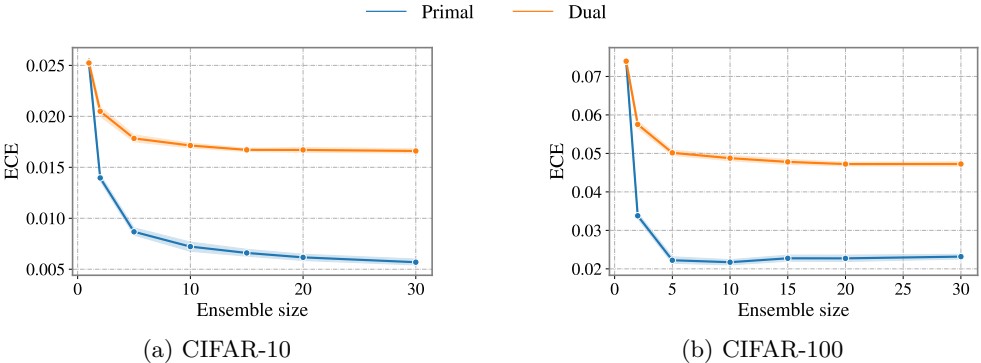

(a) CIFAR-10

(b) CIFAR-100

Figure 15: Primal and dual expected calibration error (ECE) on the Cifar10 and Cifar100 datasets.

# H    Primal and dual ensembling for $n = 2$

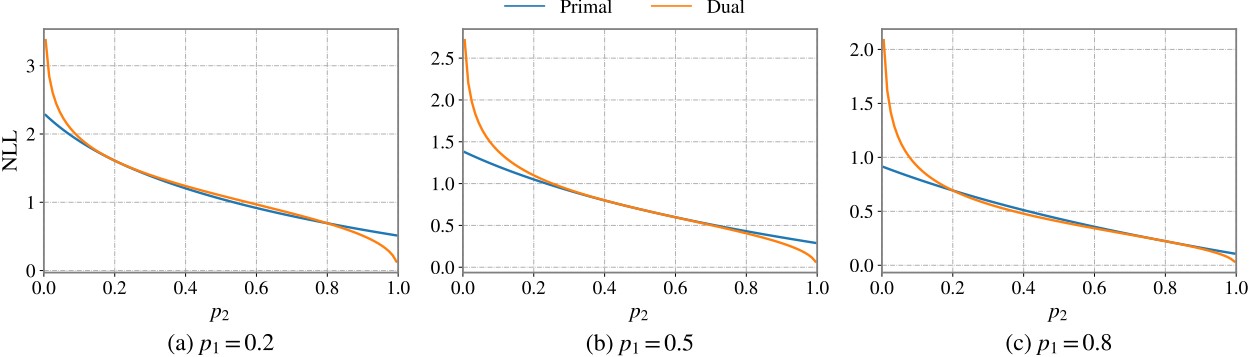

(a) $p_1 = 0.2$

(b) $p_1 = 0.5$

(c) $p_1 = 0.8$

Figure 16: Ensembling two models with the cross-entropy loss. When the second model's predicted probability for the true class $p_2$ is extreme (either towards 0 or towards 1), it dominates the prediction made by the other model $p_1$. This allows dual ensembling to achieve a lower loss when the second model is correct ($p_2 \to 1$), but also achieves much higher losses when the second model is incorrect ($p_2 \to 0$). Note that primal and dual losses are equal at $p_1 = p_2$ and $p_1 = 1 - p_2$, as predicted by Proposition 6.1.

