# OpenReview forum: "Ensembles of Classifiers: a Bias-Variance Perspective"
_TMLR — Accepted by TMLR_

### Review · Reviewer_KKeP · 2022-08-08

**Summary Of Contributions:**

This paper presents a theoretical tool to analyze the bias-variance tradeoff under general Bregman divergence and studies deep ensembles using it. Specifically, the paper first reveals the dual-reparameterization of the "central prediction" term in the generalized Bias-Variance Decomposition (BVD) and extends the law of total variance to the generic notion of variance defined under the dual parameterization. Proceeding further, the paper shows the effect of ensembling on the bias and variances terms, and show that the primal ensembling (the typical ensembling done in the function output space) may increase or decrease the bias unlike the traditional ensembles under MSE loss. Instead, the paper shows that the ensembles done in the dual space actually preserves the bias while reducing the variance as expected in the typical MSE loss. The presented theories are empirically backed up using image classification benchmark datasets.

**Requested Changes:**

- Are there more examples of the loss functions being used for deep networks that actually correspond to a special case of the presented framework? If so, it would be nice to see whether the ensembling reduces the bias as in the presented experiments with cross-entropy loss.
- As noted above, it would be good to see similar experiments on datasets other than CIFAR10 or CIFAR100; e.g., SVHN, Tiny ImageNet, or even ImageNet if affordable.
- Typically, the deep neural networks are trained with regularization terms such as weight decays. I wonder how this relates with the presented theory; does the cross entropy augmented with weight decay naturally fall into the presented framework?

**Strengths And Weaknesses:**

First of all, I must admit that I don't have a strong background in the related literature, so my assessment of the novelty may be incorrect.

Strengths
- The paper provides a novel parameterization that leads to an interesting bias-variance analysis of ensembles.
- The empirical validations, although restricted to CIFAR10 and CIFAR100, are thorough and well match the theory.
- The paper is generally well written and provides a nice summary of the related works.

Weakness
- The experiments only consider the image classification tasks trained with the cross-entropy loss, while the theory in principle covers arbitrary Bregman divergences.
- At least from the theory, it is not known under what circumstances the primal ensembling reduces the bias in the image classification tasks. Also, most of the experiments are done only for CIFAR10 and CIFAR100, so it is not clear whether this bias reduction is specific to those two datasets or more common in generic image classification tasks.
- The empirical confirmation is based on conditional biases and variances, not the original biases and variances.

---

> ### Author Response · Authors · 2022-08-19
> **Response to reviewer KKeP**
>
> Thank you for your review!  The corresponding modifications were added in red to the updated manuscript.
>
> Weaknesses:
> - Experiments with other Bregman divergences: It would be quite interesting to visualize the decomposition for other Bregman divergence losses. However, the two main losses that are used to evaluate models (not including regularization terms that are typically removed at evaluation) are the mean squared error and the cross-entropy / NLL loss. We would also like to emphasize that the major part of our contribution is theoretical; experiments in section 6.1 serve to visualize the behavior predicted by theory.
> - When does primal ensembling reduce bias / is this behavior specific to Cifar: Indeed, the bias reduction observed on the Cifar datasets should not be generalized further than our experiments. However, most figures report the average bias/variance/nll over the entire test set; the pointwise analysis reveals that the dual bias is lower on certain points, and higher on others. We have added a discussion to section 6 to reflect this. In particular, one may notice that when the loss is very low (lower left side of the scatterplots of Fig. 4a), the dual bias is lower than the primal bias. We updated Fig. 4a to include a histogram of frequencies, reflecting that this situation is actually more common than may be expected. We also added a visualization of when primal ensembling can be outperformed by dual ensembling in the two-class scenario in Appendix H.
> - Conditional/unconditional estimates: Indeed, one would ideally obtain unconditional bias and variances. However, as described at the end of section 4, the alternative lies in bootstrapped estimates that are significantly costlier; we already require at least 20n models to obtain a conditional estimate for ensembles of size n on Cifar10 (estimates for smaller numbers of draws are significantly noisier). Furthermore, we would like to insist upon the fact that -- barring some rare exceptions -- most bias-variance decompositions, and even general ML model analyses, rely on conditional estimates (or partitioned estimates on subsets of training data), putting us in line with the broader literature.
>
> Requested changes:
> - More examples of Bregman divergences in the literature: The KL divergence and MSE are by a wide margin the most used Bregman divergences in ML; we have updated the related work to include a discussion of additional (asymmetric) Bregman divergences that have been used in recent ML work. Regarding whether or not such losses would have a reduced bias for primal ensembling, we expect this will be highly loss-dependent. As our updated analysis shows (section 6.1) for the CE loss, the average bias is reduced, but on the majority of inputs, the primal bias is actually higher by a small amount. The average reduction is explained by the fact primal ensembling is less sensitive to incorrect predictions than dual ensembling.
> - Additional experiments: We have added experimental results on SVHN_cropped dataset (Appendix D); the conclusions remain the same as on the Cifar dataset (although the bias is slightly lower than on Cifar10). However, we again insist upon the fact that the plot shows the bias, variance, and NLL averaged over all test points.
> - Regarding regularization: This is a very interesting question, with two key distinctions.
>     1.  Does our analysis apply to models trained with weight decay or other regularization schemes? Our theoretical analysis is mostly agnostic to how the models are trained (with or without weight decay, data augmentation, etc.); the only assumption that is required is that when we ensemble models, these models are drawn iid from the same distribution. Thus, regularizations that apply to ensemble members jointly (e.g., to increase ensemble diversity) would violate this assumption. Regularization applied to individual models such as weight decay, however, does not affect our analysis.
>     2.  Does our theory apply when decomposing a loss that includes a weight decay term? Weight decay (the squared L2 norm of the weights) is a form of Bregman divergence (the L2 loss between the weights and the zero vector). As the sum of two Bregman divergences is also a Bregman divergence, one could apply our analysis to losses that include weight decay. However, the weight decay term is typically only used during training, and not when evaluating the model at convergence, so it is not clear if such an analysis would be insightful.

---

### Review · Reviewer_YLUi · 2022-08-08

**Summary Of Contributions:**

This paper derives a dual formulation of the bias-variance decomposition for Bregman divergences [1], and analyzes several properties of model ensembles for cases where typical assumptions (symmetry, joint convexity) do not hold. Namely, for asymmetric Bregman divergences (e.g. KL divergence), primal ensembles are distinct from dual ensembles. For classifiers trained with cross entropy loss, the latter corresponds to averaging logits rather than output probabilities. While dual ensembles keep bias fixed and reduce variance, primal ensembles may either increase or decrease bias while reducing variance. The authors verify their theory with experiments on ensembles of neural network classifiers, and offer insight as to which would perform better in practice.

[1] D. Phau. A Generalized Bias-Variance Decomposition for Bregman Divergences. 2013. http://davidpfau.com/assets/generalized_bvd_proof.pdf

**Broader Impact Concerns:**

The authors could comment on how the bias-variance decomposition for deep network ensembles relates to notions of bias, fairness, and calibration in ML systems, i.e. whether ensembles with lower statistical bias are inherently more fair.


**Requested Changes:**

### Critical
- In addition to the related work section, cite relevant work in Section 5 where the results generalize earlier analysis on primal/dual ensembles.
- Provide additional motivating examples of asymmetric Bregman divergences in ML

### Non-critical
- Comment on calibration error of primal vs dual ensembles
- Show an empirical example of a neural network primal ensemble in which bias increases with ensemble size
- Additional experiments involving other asymmetric Bregman divergences
- Typo
    - "whereas the bias unchanged" should be "whereas the bias is unchanged"


**Strengths And Weaknesses:**

### Strengths
- Natural, elegant extension of previous work on model ensembles and Bregman divergences
- Intermediate results such as the law of total variance may be of independent interest
- Well-organized with clear explanations and proofs

### Weaknesses
- This paper would be strengthened if the contributions included a setting that distinguished itself from what has been covered by previous work, e.g. analyzing another asymmetric Bregman divergence besides KL divergence, or an application in which the dual ensemble outperforms the corresponding primal ensemble
- A better conclusion would offer directions for future work, in addition to reiterating what was outlined in the introduction.

---

> ### Author Response · Authors · 2022-08-19
> **Response to reviewer YLUi**
>
> Thank you for your review! The corresponding modifications were added in green to the updated manuscript.
>
> Weaknesses:
> - Including an example where dual ensembles outperform primal ensembles. Thank you for this suggestion! We have updated section 6.1 to include a detailed discussion of when this happens, including empirical examples corresponding to the trained models on Cifar100.
> The proof of Proposition 5.2 also includes an example of when this happens in the two-class setting, and we added a visualization in Appendix H of how varying the prediction of one ensemble member can flip the ordering of primal and dual ensemble losses.
> - Other losses: As mentioned to reviewer KKeP, the two main losses that arise in ML (not including regularization terms that are typically removed at evaluation) are built upon the mean squared error and the cross-entropy / NLL loss. Although the major application of our work being (as of now) restricted to the NLL/cross-entropy loss, we would argue that the vast prevalence of this loss across the wealth of ML tasks that exist today makes our theoretical advances a valuable contribution to the field. We have added references to other use cases of asymmetric Bregman divergences in the ML literature to the related work; such divergences include the Itakura-Saito distance, the I-divergence (a generalization of the KL-divergence), and learned/parametric Bregman divergences.
> - Better conclusion: We have updated our conclusion in response to your comments.
>
> Requested changes:
>
> Critical changes:
> - Section 5: We have updated this section with the relevant references (as well as section 6); please let us know if there are specific additional references that should be included.
> - Motivating examples of asymmetric Bregman divergences: we have updated the related work, adding references to the (asymmetric) Itakura-Saito distance, which has been used for nonnegative matrix factorization tasks in audio processing, as well as recent work that leverages parametric or learned  Bregman divergences.
>
> Non-critical changes:
>
> - Calibration error: We also added a visualization of the Expected Calibration Error (ECE) on Cifar10 and Cifar100 to Appendix G. In line with the lower NLL from primal ensembles, we also find that primal ensembles tend to achieve a lower ECE than dual ensembles.
> - Empirical example of bias increase: We have updated section 6 (and Figure 4) to highlight empirical cases where primal ensembles have a higher NLL than dual ensembles; since the dual bias is constant (equal to the bias for ensemble size = 1), such points also correspond to an increase in primal bias when adding ensemble members.
> - Additional experiments: We agree that it would be interesting to extend our experimental analysis to other (asymmetric) Bregman divergences! The main scope of our paper lies in the theoretical analysis of the bias-variance decomposition, using experiments to verify and illustrate some potential applications of our analysis on the most common asymmetric Bregman divergence (the KL divergence). We have added this topic as a direction for future research.
> - Typo: fixed
> - Broader impact concerns: Thank you for bringing this up! Although the definitions of bias are different within the fairness literature, the fact that the (BVD) bias is conserved under dual ensembling may be of interest when seeking guarantees for model performance when ensembling at test time.

---

### Review · Reviewer_iTBd · 2022-08-15

**Summary Of Contributions:**

The paper proposes a justification for the success of ensemble models in various prediction tasks. The main novelty is the analysis through the generalization of the bias-variance tradeoff to Bregman divergences, thereby extending it beyond Mean Squared Error (MSE) to asymmetric losses like the KL divergence. The authors demonstrate that in this case the mean prediction from the bias-variance analysis with MSE can be replaced by the primal projection of a dual mean. Using this, the authors show that (conventional) model ensembling in the primal space reduces variance and may increase or decrease bias, while ensembling in the dual space decreases variance and leaves bias unchanged. Experimental results validate these claims though bias with primal ensembles is almost always lower than that of dual ensembles, and appears to decrease with increasing ensemble size.

**Requested Changes:**

I like the novelty of the idea and the extent of the analysis and experiments. I think the paper will be ready for publication if the following changes are made to the explanation:

1. I would recommend adding a running example in Section 3 since all readers interested in ensemble models may not be familiar with Bregman divergences. A good candidate for this would be the KL divergence. Currently Remark 3.4 mentions that if $F$ is the cross entropy loss then $D_{F]$ is the KL divergence. I would recommend using this example throughout Section 3 to illustrate the convex conjugate, the identity in (2), and the results of Proposition 3.1, 3.2, 3.3 for the cross entropy/KL divergence case.

2. Please mention the exact meaning of $X$, $Y$, and the sources of randomness over which the expectation may need to be taken, at the beginning of Section 3.2 before introducing $\mathbb{E} D[Y||X]$.

3. Since the conditional bias and variance are approimations, the expression in Propsition 4.1 should not be an equality right (LHS is the same as the LHS of (3) but the RHS is an approximation)?

4. The description of the experiments comparing conditional and bootstrapped estimates is a bit vague. Particularly the word partition seems to be used in different contexts. In the context of conditional bias, the authors say "one can estimate bias and variance by partitioning the training set into disjoint subsets", while later the authors say "these partitions allow us to compute the true bias and variance of the algorithm". It seems like partitioning the entire dataset allows computing of the true bias and variance while each partition is further partitioned into sub-partitions to compute the conditional and bootstrapped estimates. Please clarify if this is the case and please specify the number of sub-partitions (is it equal to \# trained models?) for the conditional and bootstrapped estimates in each case.

5. Related to the above point, the meaning of "partition multiple" in the plots in Appendix C was not clear to me. Please clarify.

6. Please include the relevant expressions to show how the loss of ensembles averaged in the logit space is equivalent to the ensemble diversity regularizer (Remark 5.3). This will also be easier to convey if the KL divergence running example is included in Section 3 as mentioned in point 1 above.

6. Please completely specify experimental conditions for Section 6. Currently the partition sizes, and number of partitions for each experiment are not mentioned. Can you also specify the values of bias, variance, NLL etc for the case when a single model is used? The usefulness of the dual ensemble will be quite limited if it does not outperform a single model.

7. X-axis labels are missing in Fig 4(b), 4(c), 4(d).

8. Please summarize the experimental results more concretely in the last paragraph of Section 7. The current paragraph describes the experiments conducted but does not include all results/conclusions from the experiments.

**Strengths And Weaknesses:**

Strengths:

1. Novel analysis applying the bias variance tradeoff for Bregman divergence to model ensembles
2. Introduction of a new class of (dual) ensembles whose behaviour (same bias, lower variance) is clearly justified by the theory introduced in the paper
3. Extensive experiments comparing primal and dual ensembles, and comparing primal ensembles of different model architectures which is demonstrated to be superior to using the same architecture for all models in the ensemble.

Weaknesses:

1. Explanation could be clearer (see requested changes below)
2. Experimental results (superior performance/lower bias of primal ensembles compared to dual ensembles) are not fully explained by the theory.

---

> ### Author Response · Authors · 2022-08-19
> **Response to reviewer iTBd**
>
> Thank you for your detailed review! The corresponding modifications were added in blue to the updated manuscript .
>
> Weaknesses:
>
> - Explanation could be clearer: we have made the requested changes (visible in blue in the updated manuscript).
>
> - Lower bias of primal ensembles: we have updated section 6.1 to include a more in-depth analysis of this phenomenon, which we hope provides some additional clarity; to summarize, even a single ensemble member can make the dual loss arbitrarily worse than the primal loss, but the improvements over the primal loss are bounded.
>
> Requested changes:
> - We have added the running example to section 3 and clarified the meaning of X and Y at the beginning of 3.2.
> - Conditional bias / variance: The identity is an equality. The expected loss can either be computed using the classical BVD (equation 3), or by writing the loss as $\mathbb E L(y, x) = \mathbb E_z[ \mathbb E L(y, x) | z]$, then applying the bias-variance decomposition to $\mathbb E[ L(y, x) | z]$. The latter is how we obtain the right-hand-side of Proposition 4.1.
> - Conditional/bootstrapped estimates: The full training dataset is divided into 50 partitions. The true bias and variance values use the models where each model is trained on a different partition out of the total 50 and a different random seed (using # trained models out of the total models for the estimates). To compute the conditional bias and variance values, we choose one partition out of the total 50 partitions and then 50 models are trained on the same partition chosen but with a different random seed and then the conditional bias and variance values are calculated using the # trained models. Note that there is no sub-partition in this case because conditional estimates are conditioned on the training dataset being same and thus only approximate the true bias and variance values. Here, only one partition is used as the training data set for the models. To calculate the bootstrap estimates, again the same partition is used as the training data and given to Algorithm 1 which computes the bootstrap estimates. Here, models are trained on the bootstrap samples of the partition used as the training set; we have updated the paper to clarify this.
> - Partition multiple: Partition fixed corresponds to the conditional estimates of the bias and the variance (models are trained on a fixed partition). Partition multiple corresponds to the case where each model is trained on a different partition and a different random seed and thus, would start converging to the true bias and variance values as the number of partitions increase (since this is a consistent estimator by the law of large numbers). Note that here the x-axis also corresponds to the number of trained models and the total number of partitions is fixed beforehand to 20.
> - Diversity regularizer: We have added the derivation to the end of section 5.2.
> - Summarization of results in section 7: We’ve made the requested changes.
>
> Experimental conditions for section 6:
>
> - Partitions: As mentioned in the last paragraph of section 4, all experimental results beyond the bootstrap analysis consider conditional estimates. Although a limitation, this restriction is due both to the increased cost of bootstrapped experiments (we need 20*30 draws of a single model to accurately estimate the conditional bias for ensembles of size 30 on Cifar), and to align us with the existing literature on neural network analysis.
> - Performance of size-1 ensembles: we assume you mean for Figures 5 and 6 (the size-1 ensemble performance is provided in Figures 3a and 3b). Indeed, the size-1 ensembles aren’t included for the depth/width experiments as their performance is comparatively much worse, decreasing the readability of the figures. We have added a short sentence discussing this in the caption, and added their values to Appendices E and F.

---

### Author Response · Authors · 2022-08-19
**Comment to all reviewers**

We thank all three reviewers for their careful review of our submission. We have attempted to address all questions in the updated version of the manuscript; for ease of review, we’ve highlighted all major changes in different colors based on the corresponding reviewer:
- Reviewer KKeP: Red
- Reviewer YLUi: Green.
- Reviewer iTBd: Blue.

The largest update is included in section 6.1, and consists of a more detailed analysis (theoretical and empirical) of cases where primal ensembling can be better---or worse---than dual ensembling. We would also like to note that the lower loss for primal ensembling under the CE loss has been noted in (Brofos & Shu, 2019).

---

### Decision · Action_Editors · 2022-09-16

**Recommendation:** Accept as is

**Comment:**

Thanks for your submission to TMLR.  The reviewers all felt that the paper was strong enough to be accepted, and that the authors have responded adequately to the concerns of the reviewers with their revisions.